# Multiple Copies of microRNA Binding Sites in Long 3′UTR Variants Regulate Axonal Translation

**DOI:** 10.3390/cells12020233

**Published:** 2023-01-06

**Authors:** Luba Farberov, Ariel Ionescu, Yazeed Zoabi, Guy Shapira, Amjd Ibraheem, Yosi Azan, Eran Perlson, Noam Shomron

**Affiliations:** 1Sackler Faculty of Medicine, Tel-Aviv University, Ramat-Aviv, Tel-Aviv 69978, Israel; 2Edmond J. Safra Center for Bioinformatics, Tel-Aviv University, Ramat-Aviv, Tel-Aviv 69978, Israel; 3Sagol School of Neuroscience, Tel-Aviv University, Ramat-Aviv, Tel-Aviv 69978, Israel

**Keywords:** KIF5B, miR-129-5p, axons, 3′UTR, seed, RNA localization, RNA translation

## Abstract

Rapid responses to changes within subcellular compartments of highly polarized cells, such as neuron axons, depend on local translation and post-transcriptional regulation. The mechanism by which microRNAs (miRNAs) regulate this process is not fully understood. Here, using live cell imaging and RNA sequencing analysis, we demonstrated how miRNAs can differentially control hundreds of transcripts at the subcellular level. We demonstrated that the seed match length of the miRNA target-sequence regulates both mRNA stability and protein translation rates. While longer seed matches have an increased inhibitory effect, transcriptome analysis did not reveal differences in seed match length between axonal and somata mRNAs of motor neurons. However, mRNA variants with longer 3′UTR are enriched in axons and contain multiple repeats of specific miRNA target sequences. Finally, we demonstrated that the long 3′UTR mRNA variant of the motor protein Kif5b is enriched explicitly in motor neuron axons and contains multiple sequence repeats for binding miR-129-5p. This subsequently results in the differential post-transcriptional regulation of kif5b and its synthesis in axons. Thus, we suggest that the number of miRNA binding sites at the 3′UTR of the mRNA, rather than the miRNA seed match length, regulates the axonal transcriptome.

## 1. Introduction

Motor neurons (MNs) are highly polarized cells whose most extended projection, the motor axon, extends to over one meter long in adult humans. Due to the remote distances from their cell bodies (soma), distal axons and synaptic terminals depend on two distinct processes: (i) bidirectional axonal transport, facilitated by motor proteins [1,2] and (ii) local mRNA translation [3,4,5]. The combination of these two processes enables MNs to respond efficiently to intracellular and extracellular cues crucial for their survival and function [6]. To perform local translation, MNs must shuttle mRNAs into axons [7,8]. This occurs largely via RNA-binding proteins that bind specific regulatory motifs in the mRNA untranslated regions (UTRs) and also tether motor proteins [7,8,9]. Previous studies found that mRNA transcripts localized to axons tend to be longer than those in the soma compartment, thus suggesting the possible role of these transcripts in RNA localization and regulation [10,11,12]. Axonal and synaptic localization of micro RNAs (miRNA), as well as the RNA-induced silencing complex (RISC), was previously reported in several studies [13,14,15,16]. This implies that RNA silencing could be a mode of local regulation over mRNA translation in axons and synapses. However, knowledge is still limited regarding mechanisms of translational regulation of axonal mRNAs, and specifically of the means by which miRNA regulate their targets in axons. 

MiRNAs are small, endogenous non-coding RNAs, about 21 nucleotides long, which induce posttranscriptional silencing of their target genes, thereby functioning as negative gene expression regulators [17]. miRNAs play a key role in regulating the transcriptome and proteome in distinct subcellular regions, including axons. In mammals, miRNAs are predicted to control the activity of about two-thirds of all coding genes [18,19,20]. To exert their functional regulation, the 5′ positioned 2-8 nucleotides of the miRNA (also known as the seed region) bind the target mRNA 3′ untranslated region (3′ UTR) sequence [17,20,21]. Consistent with the relatively short complementary sequence required for target gene recognition, each miRNA is estimated to target several tens to hundreds of transcripts [22]. Analysis of conserved miRNA-3′UTR pairing has led to the identification of effective canonical site types: 8mer (bases 1–8), 7mer-m8 (bases 2–8), 7mer-A1 (bases 1–7), and 6mer (bases 2–7). Longer seed regions are believed to have greater mRNA repression efficacy [23,24,25]. These canonical seeds are estimated to represent at least two-thirds of all miRNA targets. Moreover, the number of canonical sites influences target inhibition efficacy; each additional site enhances target gene repression [26]. However, how these traits directly influence protein synthesis of target transcripts, and how this mode of regulation translates into function specifically in subcellular domains such as the neuronal axons, remain mostly unknown.

Here, using a unique live cell imaging approach, we characterized the regulatory influence of variations in miRNA:mRNA seed matching on target mRNA levels and on the mRNA translation rate. We revealed that the rate of protein synthesis is decreased in correlation to target complementation. Furthermore, we demonstrated a key mechanism in the regulation of axonal transcriptome, by which multiple repeats of miRNA target sites in the long 3′UTR of axonal mRNA variants, rather than variations in seed matching length, dominate the translation of transcripts. 

## 2. Materials and Methods

### 2.1. Animals

HB9-GFP (Jax stock no. 005029) mice were originally obtained from Jackson Laboratories. The colony was maintained by breeding with ICR mice (Institute of Animal Science, Harlan Biotech, Rehovot, Israel). All the animal experiments were approved and supervised by the Animal Ethics Committee of Tel-Aviv University (Ramat-Aviv, Tel-Aviv, Israel).

### 2.2. Cell Culture

Monolayer-adherent HEK-293T cells (transformed human embryonic kidney cells) or SH-S5Y5 (human neuroblastoma cells) were grown in Dulbecco’s modified Eagle’s medium (DMEM) (Thermo Fisher Scientific, Waltham, MA, USA) supplemented with 10% (*vol*/*vol*) fetal bovine serum (FBS) (Thermo Fisher Scientific, Waltham, MA, USA), 0.3 g/liter L-glutamine, 100 units/mL penicillin, and 100 units/mL streptomycin (Biological Industries, Israel). Cells were purchased from the American Type Culture Collection (ATCC).

Cells were incubated at 37 °C in a 5% CO_2_ atmosphere. Before use, each cell line was confirmed to have no mycoplasma contamination using the EZ-PCR Mycoplasma Test Kit (Biological Industries, Beit-Ha’Emek, Israel). Prior to each experiment, the cells were stained with trypan blue and counted using the Countess automated cell counter (Thermo Fisher Scientific, Waltham, MA, USA).

### 2.3. Preparation of the Microfluidic Chamber (MFC) 

Highly pure axonal RNA was isolated by using a novel radial microfluidic device fabricated according to the SU-8 photoresist protocol in the Tel-Aviv University Nano and Micro Fabrication Center. Polydimethylsiloxane (PDMS) mold was pre-treated with chlorotrimethylsilane (Merck-Sigma-Aldrich, St. Louis, MO, USA) prior to PDMS casting. PDMS base and curing agent were mixed in a ratio of 10:1. PDMS solution was mixed until it became homogenous, cast into radial microfluid chamber wafers, incubated for 2 h within a vacuum desiccator to discard trapped air bubbles, and finally baked for 3 h at 70 °C. Radial microfluidic chambers (MFCs) were then punched twice to form MFC rings. The inner well was punched with a 7 mm biopsy punch, and the outer well with a 9 mm punch. The cleaning procedure was done in a similar manner as for regular MFC. Radial MFC rings adhered to sterile 13 mm coverslips inside 24-well plates.

### 2.4. Primary Motor Neuron Cultures

Ventral spinal cords from E12.5 embryos were dissected in Hanks’ Balanced Salt Solution (HBSS) prior to dissociation. Dissociated MN cultures were obtained by trypsinization and trituration of explants. The upernatant was collected and centrifuged through bovine serum albumin (BSA) (Sigma) cushion. The pellet was then resuspended and centrifuged through an Optiprep (Sigma) gradient (containing 10.4% Optiprep, 10 mM Tricine, 4% *w*/*v* glucose). An MN-enriched fraction was collected from the interphase and centrifuged through a BSA cushion. The MN pellet was resuspended and plated in the central well of the radial MFC at a concentration of 250,000 MNs per chamber in 40 µL of complete neurobasal (CNB) medium, containing neurobasal, 4% B27, 2% horse serum (Biological Industries, Beit-Ha’Emek, Israel), 1% Glutamax, 1% P/S, 25 µM beta-mercapto ethanol, 25 ng/mL BDNF, 1 ng/mL GDNF (Alomone, Jerusalem, Israel), and 0.5 ng/mL CNTF (Alomone). MNs were maintained in CNB medium, which was refreshed every other day. Glial cell proliferation was restricted by the addition of 1 µM cytosine arabinoside (ARA-C; Sigma) to the culture medium in 1-3DIV. At 3DIV, the BDNF concentration in the proximal compartment was reduced (1 ng/mL), while the medium in the peripheral (axonal) compartment was enriched with GDNF and BDNF (25 ng/mL) to direct axonal growth. 

### 2.5. RNA Extraction from MN Axons and Cell Bodies

MN axonal RNA was extracted from the outer compartment of radial MFCs at 14DIV. Axonal RNA was extracted by removing the PBS (from the prior wash) from the outer compartment and adding 100 µL Qiazol lysis reagent (Merck-Sigma–Aldrich, St. Louis, MO, USA). The inner well was filled with a higher volume of PBS to disable the inward flow of the lysis reagent toward the inner (soma) compartment and to prevent soma contamination. Axons were washed off the plate by pipetting the Qiazol reagent around the outer well for 30 s. RNA from somata in the inner compartment was extracted with 100 µL Qiazol reagent, and lysate was collected in a similar manner. RNA was isolated and purified using miRNeasy micro kit (Qiagen).

### 2.6. Plasmid Construction

Destabilized GFP was generated as described by Li et al. [27]. To insert the 3′UTR counting full complementation miR-1-3p binding site, we used the QuikChange Lightning site-directed mutagenesis (SDM) kit (Agilent, Santa Clara, CA, USA). This kit was also used to create various miRNA canonical sites. The new vectors were verified by Sanger sequencing. Canonical site sequences are listed in Table 1.

### 2.7. Mature miRNA Mimics

Mature miRNAs were ordered from IDT (Integrated Device Technology, San Jose, CA, USA), according to the miRBase, a miRNA database [28].

All the miRNAs were conjugated to CY3 fluorescent dye at the 3′ end.

Mature miRNA sequences are listed in Table 1.

### 2.8. Plasmid Transfection

HEK-293T cells were seeded in a pre-coated (0.01% poly-L-lysine for 30 min) 6 mm PDMS, and well adhered to a 35 mm glass-bottom dish (API) at a concentration of 8 × 10^3^ cells/well. Twenty-four hours later, at ~60% confluence, the cells were transfected with 35 ng plasmid, or co-transfected with 33 ng plasmid and 1ng mature miRNA, using Lipofectamine 2000 transfection reagent (Thermo Fisher Scientific, Waltham, MA, USA) in accordance with the manufacturer’s instructions. S5Y5 cells were seeded in 24-well plates at a concentration of 1 × 10^5^ cells/well. For miRNA overexpression (OE), transfection was performed with 500 ng of miRVec containing the desired pre-miRNA or a control vector. 

### 2.9. qRT-PCR Analysis

Extracted total RNA was used as input for mRNA complementary deoxyribonucleic acid (cDNA) synthesis. Reverse transcription (RT) of mRNA was conducted using the random-primer and high-capacity cDNA Reverse Transcription Kit (Thermo Fisher Scientific, Waltham, MA, USA). mRNA expression was assessed using the SYBR Green Fast PCR Master Mix (Quantabio, Beverly, MA, USA), according to the manufacturer’s instructions. miRNA expression was assessed similarly, using the TaqMan Universal PCR Master Mix (Thermo Fisher Scientific, Waltham, MA, USA), in accordance with the manufacturer’s instructions.

Expression values were calculated based on the comparative cycle threshold (Ct) method [29]. mRNA expression levels were normalized to glyceraldehyde 3-phosphate dehydrogenase (GAPDH) as an endogenous control. Specific primers for mRNA expression detection were ordered from IDT (Table 2). GAPDH was chosen as a control due to its relatively stable expression [30]. miRNA expression was normalized to miR-124a-3p or U6 snRNA expression (Table 3). PCR amplification and reading were conducted on the StepOnePlus Real-Time PCR Systems (Thermo Fisher Scientific, Waltham, MA, USA).

### 2.10. Live Cell Confocal Imaging and Photobleaching

Live imaging of GFP Fluorescent Recovery After Photobleaching (FRAP) in HEK-293T cells was performed with a Nikon-Ti microscope (X40 oil objective) paired with the Andor Frappa device, Yokogawa CSU-X spinning disk, and an iXon897 EMCCD camera. FRAP on complete cells was done using a 488nm laser at 15 mW power. Dual color images of GFP and Cy3 miRNAs were acquired prior to FRAP, and for 20 consecutive times afterward at 30-s intervals. Live imaging was performed in a controlled CO_2_ and 37 °C chamber. Image analysis was performed with FIJI.

### 2.11. Live Cell Imaging FRAP Analysis of GFP Signal Build-up

The analysis of GFP signal build-up after photobleaching was performed with FIJI, using ‘Time series analyzer V3′ plugin. Briefly, the perimeter of the cell prior to photobleaching based on its GFP signal was marked as an ROI. The intensity of the GFP fluorescence was extracted from each time point. The signal build-up was calculated by subtracting the post-bleach fluorescence intensity value of each time point from the initial intensity, generating a normalized cumulative value for GFP build-up. The slope of dGFP signal build-up was calculated for the linear fraction of the curve (frames 5–20).

### 2.12. RNA Extraction

At time 0, 2, or 4 h post miR-1-3p or control RNA sequence OE, total RNA was extracted from cells using TRIzol reagent (Thermo Fisher Scientific, Waltham, MA, USA), in accordance with the manufacturer’s instructions. Final RNA concentration and purity were measured using a NanoDrop ND-1000 spectrophotometer (Thermo Fisher Scientific, Waltham, MA, USA).

### 2.13. miRNA Constructs

Pre-miRNAs were cloned into the *BamHI–EcoRI* restriction site of the miRNA expression vector, miRVec, which was provided by Prof. Reuven Agami [31]. The genomic loci of ~70 bp upstream and downstream of the pre-miRNAs were inserted into the vector.

### 2.14. Dual Luciferase Assay

Fragments of ~200 bp of Kif5b 3′UTR spanning the miRNA binding sites were cloned downstream to the Renilla luciferase reporter of the psiCHECK-2 plasmid (Promega, Madison, WI, USA) that contains a Firefly luciferase reporter (used as a control). The miRNA-binding sites were mutated by the QuikChange Lightning Site-Directed Mutagenesis Kit (Agilent, Santa Clara, CA, USA, USA). For luciferase assays, 293T cells were transfected using Lipofectamine 2000 transfection reagent, with 5 ng psiCHECK-2 plasmid containing the desired 3′ UTR, with or without site-directed mutations; and 485 ng miRVec containing the desired pre-miRNA or a control vector. Forty-eight hours after transfection, Firefly and Renilla luciferase activities were measured using the Dual-Luciferase Reporter Assay System kit (Promega, Madison, WI, USA) and LUMIstar Omega Luminometer (BMG LabTech, Ortenberg, Germany), according to Promega’s instructions. 

### 2.15. RNA-seq and Analysis

RNA quality was assessed using Agilent’s 2100 Bioanalyzer (Agilent, Santa Clara, CA, USA). Samples with an RNA integrity number (RIN) >8 were considered high quality. High-quality total RNA, 1 µg, was used for ribosomal RNA (rRNA) depletion, by means of the RiboZero rRNA Removal Kit (Illumina, San Diego, CA, USA), and stranded RNA-seq libraries were constructed using a TruSeq stranded total RNA library prep kit (Illumina, San Diego, CA, USA). Then, 100 bp paired-end sequencing was performed on an Illumina Novaseq 6000 (Illumina, San Diego, CA, USA) at Macrogene corporation (Macrogene, Seoul, Republic of Korea). Library insert sizes ranged from approximately 50 to 300 bp, with an average size of 101 bp. The quality of the raw reads was assessed using FastQC software [32]. For the DE analysis, reads were pre-processed using the ea-utils fast-mcf tool (expressionanalysis.github.io/ea-utils/, accessed on 30 October 2019), which detects and removes adaptors and poor quality bases (Qpherd < 30) at the end of the reads [33]. Reads were mapped to the human genome reference sequence GRCh37 (hg19) and transcriptome (GTF transcript annotation retrieved from UCSC) [34] using the TopHat2 RNA-seq alignment tool [35]. Gene counts were estimated using the HTSeq tool [36].

Samples were classified as miR-1-3p or a control RNA sequence OE, with three biological replicates at each group and at each time point (0, 2, or 4 h). Normalization and differential expression were performed using the DESeq2 package in the R statistical programming environment [37,38]. 

Raw data can be accessed via the NCBI GEO repository using accession number GSE188770.

### 2.16. DaPars Analysis

The bioinformatics algorithm DaPars (Dynamitic analysis of Alternative Poly-Adenylation from RNA-seq) [39], which uses a regression model to locate endpoints of alternative polyadenylation sites, was used to identify differences in alternative polyadenylation events between ALS samples and controls. To identify significant alternative polyadenylation (APA) events, we used the following cutoffs for the DaPars software: FDR < 0.05, |ΔPDUI| ≥ 0.2, and |dPDUI| ≥ 0.2. Accordingly, we analyzed differences in lengths of 3′UTR in mRNAs, between axons and cell-bodies.

### 2.17. Somatic Variant Calling

GATK4.2 was used to call short somatic variants with Mutect, using default filtering parameters to avoid low-confidence calls [40]. Using TargetScan, we filtered only length-modifying variants that overlap 3′UTR miR binding sites.

### 2.18. Timespan Differential Gene Expression

Raw sequences trimmed using fastp 0.19.6 [41] were aligned to the reference genome using STAR 2.7.3a [42]. The raw gene expression data were processed using DESeq2 1.24.0 [43], and miRNAs targeting differentially expressed genes were obtained from TargetScan.

### 2.19. Western Blot

MN-like cells (NSC-34) were cultured and transfected with either scrambled miRVec control or with miR-129-5p miRVec plasmids. Seventy-two hours after transfection, cells were lysed with a lysis solution containing PBS, 1% Triton X-100, and 1X protease inhibitor mix (Roche). Thirty µg protein lysate was loaded to SDS-PAGE and then blotted onto a nitrocellulose membrane. The membrane was blocked for 1 hour with 5% skim milk, and then incubated with mouse anti-Kif5b (kinesin heavy chain) antibody (MAB1614, Milipore; 1:500) in 5% skim milk overnight at 4 °C. The membrane was then incubated at room temperature for 1 hour with horseradish peroxidase (HRP)-anti-mouse antibody (Jackson; 1:10,000) in 5% skim milk. Rabbit anti-ERK1/2 (Abcam; 1:10,000) and HRP-anti-rabbit (Jackson; 1:10,000) were used as a loading control. 

### 2.20. Statistics

The data are presented as means ± standard error of the mean (SEM). P-values were calculated using either ordinary ANOVA with post hoc multiple comparisons, with one/two-tail Student’s t-test, or chi-square distribution, with *p* < 0.05 considered significant (* < 0.05, ** < 0.01, *** < 0.001, **** < 0.0001). Table 4, Table 5 and Table 6 show descriptive endpoint statistics for the GFP-signal build-up plots presented in Figure 1D, Figure 2C and Figure 3C.

## 3. Results

### 3.1. miRNA-Target Complementation Determines the Rate of Translation

Extending the miRNA binding site on its target mRNA, from 6–8 nucleotides, is known to exert a gradually increasing regulatory effect. To record the relative timescale of these events, we generated several miR-1 canonical sites (of its 3p sequence) and inserted them into the 3′UTR of short half-life enhanced green fluorescent protein (destabilized EGFP, dGFP) mRNA (Table 1). This rapid generation and breakdown of the GFP reporter protein, which was previously reported to possess a half-life of ~2 h [27], enabled detecting changes in protein expression in real-time using live cell imaging. We modified the 3′UTR of GFP several times to include a single binding site for miR-1-3p, each time with a different affinity to miR-1-3p: (i) full complementation (wild-type (WT) 22 nucleotides long), (ii) 8mer (binding of nucleotides 1-8 between the miRNA and the 3′UTR), (iii) 7mer-8m, (iv) 7mer-1A, and (v) 6mer (Figure 1A). Each plasmid was co-transfected with mature Cy3-miR-1-3p into 293T cells (Appendix A). We determined that 293T cells do not endogenously express miR-1 or any other miRNA of the miR-1 family, and also confirmed the potency of miR-1-3p to silence GFP mRNA 24 h after transfection (using real-time quantitative polymerase chain reaction (RT-qPCR); Appendix A–C). At 24 h post-transfection, the total GFP fluorescence was photobleached, and build-up of new GFP fluorescent signal was recorded for 10 min (Figure 1B and Appendix A). To validate that build-up of fluorescent signals originates from synthesis of new GFPs, we repeated this experiment in the presence of protein synthesis inhibitors; we did not detect any increase in the fluorescent signal following photobleaching (Appendix A–C). Notwithstanding, we detected that GFP fluorescence reappears in cells carrying any of the 3′UTR variants despite the presence of miR-1-3p (Figure 1C). Further analysis revealed that the rates of GFP build-up are ordered according to the 3′UTR target complementation to miR-1-3p: from full complementation (WT), which showed the slowest recovery, followed by 8mer complementation, then 7mer-1A, 7mer-8m, and lastly the 6mer, which appeared most rapidly after photobleaching (Figure 1D–E). Co-expression with miR-124-Cy3 or expression of only GFP (miR-1-3p 3′UTR / no miR) was used as non-targeting controls. To determine whether the differences observed in silencing kinetics (and the subsequent differences in translation kinetics) represent a global phenomenon of binding sites, we compared the relative transcript availability of all known single-sited miR-1-3p targets at 0, 2, or 4 h after transfecting miR-1-3p (or transfection-only control) into 293T cells, using next-generation sequencing (NGS) analysis of RNA transcripts (see the Materials and Methods). We observed a complementary trend in the abundance of target transcripts. Specifically, those with stronger seed-target complementation to miR-1-3p degraded more rapidly, reducing the number of available transcripts for translation at any given time (Figure 1F). These results align well with the predicted efficiency of inhibition by the various canonical binding sites [17]. We validated the RNA-seq results by RT-qPCR for three chosen miR-1-3p gene targets at each canonical site group: 8mer (SRSF9 [44], Sox9 [45], CDK6 [46]), 7mer-8m (CNN3 [47], PDCD10 [48], CDK9 [49]), 7mer-1A (GCLC [50], KLF4 [51], and CALM1 [52]) (Figure 1G). Next, we sought to determine whether the differences in rates of protein synthesis also apply to another member of the miR-1 family, namely miR-206. These miRNA share an identical sequence only at their seed region [53,54] (Figure 2A). OE of mature miR-206, with two GFP 3′UTR constructs: (i) miR-206 3′UTR full complementation (WT), (ii) miR-206 3′UTR 8mer or miR-1-Cy3 (which is another type of 8mer) resulted in a GFP build-up pattern similar to that of miR-1-3p (Figure 2B–D). Here too, we used mature miR-124 and GFP only (miR-206 3′UTR / no miR) as non-targeting controls, with no specific regulation. The results show that variations in miRNA-target complementation differentially regulate protein synthesis.

### 3.2. Synthetic miRNA Containing Seed Modifications Differentially Regulate Protein Synthesis

Next, we examined the reciprocal inhibition effect by co-transfecting five modified miR-1-Cy3 with various binding affinities to the GFP 3′UTR construct (miR-1 3′UTR plasmid). The five miR-1-Cy3 canonical sites were: full complementation (WT), 8mer, 7mer-8m, 7mer-1A, and 6mer (Figure 3A). We found that expression levels of a single mRNA variant can be manipulated differentially by altering the seed-target match strength using modified versions of miR-1-3p (Figure 3B,C). Synthesis rates of GFP were the most affected, as expected by the WT miR-1-Cy3 (full complementation); a more accelerated build-up of GFP was observed as follows: 8mer-Cy3 > 7mer-Cy3 > 6mer-Cy3 (Figure 3D). Notably, although mRNA transcripts with a single 6mer site are scarce within cells, a 6mer modified miRNA-mimetic can be used to finely tune transcript availability and the consequent translation of the gene. Accordingly, we showed that the length of the miRNA binding site can regulate both the dynamics of protein synthesis events and RNA stability. Yet how and whether this mode of regulation participates in controlling subcellular transcriptome is still unclear. In a neuronal context, can varying 3′UTR sequences or lengths be a viable mechanism by which axons regulate transcript levels?

### 3.3. Axonal Compared to Cell-Body Transcripts Have Longer 3′UTR Variants Containing Multiple Copies of miRNA Target Sequences 

MNs can be grossly divided into two distinct structural compartments: (i) cell bodies and dendrites; and (ii) axons. We previously performed NGS on RNA isolated from primary MN cultures plated on top of a 1 µm trans-well insert as a physical barrier between cell body and axonal compartments [7] (Figure 4A). RNA was extracted exclusively from each compartment and sequenced. To examine how post-transcriptional regulation is performed in MN axons, we reanalyzed the NGS data from Rotem et al. [7]. This aimed to identify sequence modifications in miRNA binding sites within 3′UTR of axonal variants. First, based on alternative splicing and nucleotide substitution, we hypothesized that axonal mRNAs contain canonical miRNA binding sites that differ from their soma variants; we performed variant-calling analysis to examine such events. Notably, this analysis identified only as few as 17 axonal transcripts with 3′UTR variants that contained a modification within a miRNA target site (Appendix A). This indicates that miRNA binding properties and target sequence length remain unchanged between axonal and somatic compartments. However, applying the DaPars tool [39] showed that for over 450 mRNAs, the frequency is significantly higher of long than short 3′UTR variants; the latter are more abundant in cell bodies (Figure 4B). This finding is in agreement with studies that identified longer 3′UTR variants in axons [10,11,12,55]. Longer 3′UTR could potentially contain more miRNA binding sites. This raises the question as to whether the number of miRNA binding sites, rather than their seed length, could be a mode of post-transcriptional regulation in axons. Thus, we compared the number of predicted miRNA binding sites within the long 3′UTR transcripts for 23 neuronal miRNAs and 20 non-neuronal miRNAs (Figure 4C and Appendix A). Indeed, we discovered that transcripts with longer axonal 3′UTRs contain more binding sites for most of the neuronal miRNAs tested. Interestingly, 189 of the 477 transcripts with long 3′UTR are predicted targets of miR-129-5p; 81 of them have more than one binding site, and 39 have more than two binding sites (Figure 4C). Thus, our findings suggest that 3′UTR length could potentially mediate post-transcriptional regulation of axonal mRNAs by containing more binding regions for specific miRNAs. Yet whether the additional miRNA binding sites are truly located in the axon-specific long 3′UTR regions remains to be determined.

### 3.4. miR-129-5p Differentially Regulates Axonal Versus Cell-Body mRNA Expression of Kif5b mRNAs

The DaPars analysis identified that the long 3′UTR variant of two miR-129-5p targets, *kif5b* and *Mapre1*, is highly enriched in axons compared to somata (Figure 4B). Furthermore, analysis of the number of seeds predicted that both targets have three binding sites for miR-129-5p (Figure 4D). The mRNA of Kif5b, which is known to mediate anterograde axonal transport [56,57], has three miR-129-5p 7mer-1A canonical binding sites, in positions: (i) 637–643, (ii) 1225–1231, and (iii) 1498–1504. Analysis of the 3′UTR coverage in axons compared to soma revealed that only the axonal, long 3′UTR variant of kif5b mRNA contains all three positions for miR-129-5p (Figure 4D). In contrast, the short 3′UTR variant, which is highly abundant in cell bodies of MNs, contains only a single position for the miR-129-5p binding site (position 1225–1231). Similarly, the 3′UTR of Mapre1, which encodes the microtubule end-binding protein 1 (EB1), is significantly longer in axons, and has three canonical binding sites for miR-129-5p, in positions: (i) 111-117 (7mer-A1) (ii) 1425–1431 (7mer-m8), and (iii) 5278–5284 (7mer-A1). Here too, we identified that only the long, axonal 3′UTR contains all three sites, one of which has a different seed sequence; whereas the short cell-body variant contains only one site (Figure 4E). Notably, the trans-well inserts used by Rotem et al. for isolating axonal RNA could be contaminated with RNA from dendrites. Thus, we verified our RNA seq findings by extracting RNA from MN axons or cell bodies grown in radial MFCs that enable isolation of highly purified axonal RNA [58], and quantified Kif5b and miR-129-5p expression. Both Kif5b mRNA and mature miR-129-5p are found in axons, although as expected, in lower amounts than in cell bodies (Figure 4F,G). However, using specific primers designed to target either the long or the short 3′UTR variants of Kif5b mRNA, we were able to distinguish and compare the expression levels of the two variants. Indeed, the long variant was highly abundant in axons, while the short variant was more prevalent in cell bodies (Figure 4H and Appendix A). Thus, Kif5b might be more tightly regulated in axons, as in this compartment, the long 3′UTR variant consisting of three miR-129-5p binding sites is much more than the short, single-site variant. 

To experimentally confirm the predicted functional interaction between miR-129-5p and Kif5b mRNA, we examined gene expression after miR-129-5p OE in SH-SY5Y cells using qPCR. We revealed that OE of miR-129-5p for 24 h (3.92-fold higher miR-129-5p) yielded a 0.26-fold decrease in Kif5b mRNA expression (Figure 5A). To establish the direct interaction and binding of miR-129-5p to Kif5B, we cloned either a portion of the Kif5b 3′UTR target-gene (WT) or a non-targeted mutant into a luciferase reporter assay plasmid, and co-transfected it together with a miRNA-129-5p vector plasmid into 293T cells (Figure 5B). Measurement of Renilla luciferase and Firefly luciferase expression 48 h following miR-129-5p OE revealed 0.33-fold lower luciferase activity in the WT Kif5b 3′UTR than in the mutant (Figure 4C). Additionally, we identified 60% reduction in endogenous protein expression of Kif5b in mouse MN-like cell lines (NSC-34) following 72 h of miR-129-5p or scrambled-control OE (Figure 5D,E). Hence, we conclude that Kif5b is a valid target of miR-129-5p, which can differentially control the expression of Kif5b variants according to their spatial distribution. Mature miR-129-5p can be found in both axons and cell bodies, though its quantity does not predict inhibition efficiency. Therefore, the number of miRNA binding sites is probably indicative of potential regulation of its target expression. 

### 3.5. The Multiplicity of miR-129-5p Sites in the Long 3′UTR Variant of Kif5b Slows Its Translation Rate

To confirm our speculation that the number of miR-129-5p binding sites within Kif5b 3′UTR is associated with its rate of translation, we returned to our GFP platform. We inserted either the long or short 3′UTR of Kif5b into the dGFP construct and transfected the variant into 293T cells (Figure 6A). Since 293T cells endogenously express miRNA-129-5p (Appendix A), we chose not to overexpress miR-129-Cy3. Indeed, testing the clean miRNA-target interaction with a full 3′UTR containing multiple miRNA target and various other regulatory sequences is complex. Nonetheless, the GFP real-time reporter system predicted expression differences between the short and long 3′UTR variants of Kif5b (Figure 4B–D). Next, to illustrate that the effect of the miR-129-5p site number on the translation rate was “cleaner”, we inserted one, two, or three miR-129-5p binding sites into the 3′UTR of GFP; and similarly transfected each into 293T cells (Figure 6E). Analysis of the rate of fluorescent build-up after photobleaching 24 h following transfection revealed that protein synthesis rates correlated with the number of miR-129-5p 7mer-1A canonical sites: 1 site > 2 sites > 3 sites (Figure 6F–H). Hence, both approaches revealed a positive association between the number of miR-129-5p canonical sites and miRNA’s ability to downregulate its target mRNA expression. Finally, we reanalyzed the NGS data acquired from several time points after miR-1-3p transfection into 293T cells (in Figure 1). This aimed to characterize the global kinetics of endogenous target downregulation depending on the number of miRNA binding sites. These analyses, for all the miR-1-3p predicted targets, as well as specifically for the 7mer-A1 and 8mer targets, confirmed the global relation between the number of miRNA sites and the fidelity of target downregulation. This relation eventually determines transcript availability (Figure 6I–K). Lastly, we attempted to insert miR-129-5p into somata and axons of primary motor neurons grown for 14DIV in radial microfluidic devices. RNA was isolated from each compartment at 0 and 4 h post-transfection, and the mRNA levels of kif5b were evaluated using qPCR (Appendix A). This highlighted that axonal kif5b transcripts are more susceptible to miR-129-5p decay than in cell bodies (Appendix A). Notwithstanding, the contribution of long 3′UTR variants, and specifically the multiplicity of miRNA binding sites, to the stability of axonal mRNA needs to be further characterized.

Altogether, we introduced a novel method for studying miRNA regulation on protein expression in real time. This enabled the discovery that both seed complementation and the number of miRNA sites can participate in post-transcriptional cellular regulation. However, our data imply that the number of miRNA binding sites within longer 3′UTR variants is the predominant means by which miRNAs achieve differential post-transcriptional regulation in MN axons (Figure 7). These findings help elucidate mechanisms by which cells regulate their local gene expression levels.

## 4. Discussion

Axonal mRNA translation is a key process that has been shown to be critical for diverse neuronal functions, including growth and development [9,59,60,61], synaptic plasticity [5,62,63,64], response to injury [65,66], and local support of mitochondria [58,67,68]. Previous studies, including our own, have revealed that axons and cell bodies share a large portion of their transcriptome [7]. However, it is not completely clear how these two distinct compartments differentially regulate their local proteome. As master post-transcription regulators of gene expression, miRNAs have been repeatedly reported in axons [17]. Nevertheless, the mechanism by which miRNA coordinates complex processes with high specificity and fidelity at the subcellular level is still not fully understood. 

Here, by using a microscopy-based method to monitor protein synthesis rates in real-time, we revealed how modifications in the binding properties of miRNA to their target UTRs can differentially regulate transcript levels, and consequently the rate of miRNA translation. This sharpens the notion that miRNAs do not function according to the ‘all or none’ principle, but rather have a fine-tuning mode that enables dynamic control over the stability and translation of their targets. This fine-tuning mode is regulated via the number of miRNA binding sites and the type of target site. This complex level of combinatorial regulation can potentially dictate the required stoichiometry of a set of proteins for performing a specific process or function at a specific time point. Comparing NGS data from primary MNs, we found that axonal 3′UTRs of hundreds of transcripts were longer than the 3′UTRs of their corresponding transcripts in cell bodies. We showed that these long 3′UTRs contain more binding site copies for neuronal miRNAs. Specifically, miR-129-5p is predicted to target almost 40% of the transcripts with significantly longer UTRs in axons, and 17% of those with more than one binding site. We validated our findings on Kif5b, a predicted target of miR-129-5p, with three binding sites in the 3′UTR of its axonal variant, and with one of the transcripts with the longest 3′UTR in axons compared to cell bodies. We showed that in MN axons, the long 3′UTR variant of Kif5b is more abundant than the short 3′UTR variant, and contains all three binding sites for miR-129-5p. In contrast, cell bodies contain more of the short variant, with only a single miR-129-5p site. These findings imply that the long 3′UTR variant is promoted into axons, where it might also be stabilized through binding of various RNA-binding proteins, specifically to the unique 3′UTR regions. Lastly, after verifying that kif5b is a valid target of miR-129-5p, we demonstrated that the length of its UTR can determine its translation rate. This, we showed, is dependent on the number of miR-129-5p binding sites. 

Taken together, our findings describe two mechanisms by which regulating gene expression by miRNAs is finely-tuned due to the various properties of their cognate 3′UTRs. Namely, seed-target sequence complementation, variations in the number of binding sites, or both, can determine the silencing efficacy of a single miRNA. The first mechanism seems to vary more between genes. However, the second mechanism may also apply within variants of the same gene, and thus achieve discrete regulation of subcellular transcriptomes, which is fundamental for highly polarized cells such as MNs.

In addition, for the purpose of reciprocating target complementation experiments, we designed modified miRNA mimetics. Using these, we demonstrated that the ability to fine-tune synthesis rates of a specific protein is not limited to the target sequence within the protein’s 3′UTR but can also be affected by modifying the synthetic miRNA itself. 

Our findings indicate that in axons, the long 3′UTR variants are more abundant than the short ones, which may contradict the hypothesis that long 3′UTR sensitizes the transcript for stronger degradation. However, our observations were taken at a single timepoint during the development of MNs. Thus, we cannot rule out the possibility that during this time, Kif5b mRNA in axons was bound and sequestered by axonal RNA-binding proteins [10], and that its silencing by miR-129-5p could occur only after its release for translation. Additionally, levels of miRNAs can change dramatically along the course of cellular differentiation and maturation, and also following uptake from neighboring cells. Hence, the spatiotemporal expression of miRNA-129-5p can be critical for controlling various processes at later phases of MN maturation and in a subcellular manner (e.g., synaptogenesis). Moreover, axons and synapses are strictly reliant on the spatial and temporal proteome, and unregulated synthesis and accumulation of proteins within axons and synapses could have a direct impact on their function and health. The combination of spatially localized axonal miRNAs together with more susceptible mRNAs, can perhaps enable the spatiotemporal distribution of mRNAs across the axonal sub-compartments, as well as restrict the random spreading of transcripts throughout the cell. 

Aberrant miRNA expression [7,69,70] and deficits in RNA processing and translation [4,68,71], and also defective axonal transport [2,72,73] and mitochondrial abnormalities [74], are mechanisms associated with MN degeneration in amyotrophic lateral sclerosis (ALS). Interestingly, several recent works have specifically identified miR-129-5p as a new ALS-linked miRNA and suggest it confers protective traits [75,76]. Intriguingly, our unbiased approach identified miR-129-5p as a potential master regulator of axonal proteome and highlights the motor protein Kif5b and the microtubule-associated Mapre1 as two of its ‘multiple binding site’ targets in axons. Kif5b mediates the anterograde axonal transport of diverse organelles, including mitochondria, into distal axons and synapses [56,57,77]. Recently, the transcripts of several kinesin superfamily motor proteins were identified in distal axons and presynaptic terminals of excitatory neurons. This suggests that these are locally translated to efficiently transport RNA granules, newly synthesized cargo, and mitochondria into the synapse [5]. Specifically, mutations in the Kif5A gene, a closely related kinesin, are causes of various neurological disorders including Charcot-Marie Tooth and ALS [73,78,79]. Therefore, our findings potentially link the critical deficits in ALS and suggest that miRNA-129-5p may regulate the axonal local synthesis of Kif5b, which consequently participates in the vital process of axonal transport. Future research should investigate the specific involvement of miR-129-5p in ALS pathogenesis, and attempt to unveil its axonal targets and the local processes it regulates.

To summarize, our findings illustrate several mechanisms that cells have adapted to achieve an efficient, versatile, yet potent response to the expression of a single miRNA. These involve changes in target complementation and the number of miRNA binding sites; and distinctly distributing transcript variants to subcellular compartments. 

## Figures and Tables

**Figure 1 cells-12-00233-f001:**
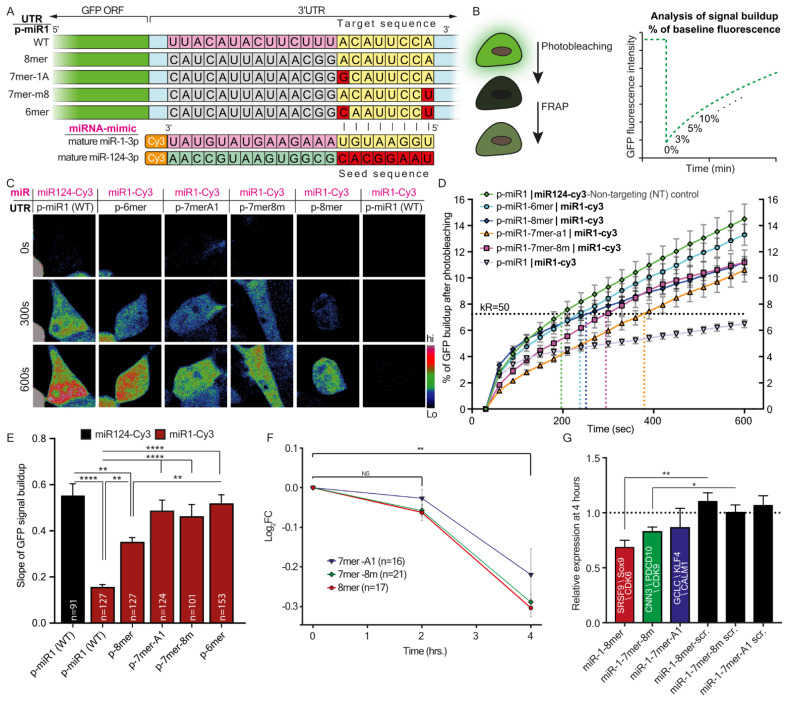
Target GFP downregulation caused by a number of miR-1-3p canonical sites located on the target 3′UTR. (**A**) Schematic illustration of the experimental setup. Five dGFP expression plasmids containing miR-1-3p canonical sites in the 3′UTR of dGFP were each co-transfected together with mature Cy3-miR-1, or with mature Cy3-miR-124 as a control, into 293T cells. Green indicates the GFP coding sequence (CDS); light blue indicates the 3′UTR of dGFP; nucleotides labeled in pink indicate nucleotides in full complementation to mature miRNA; nucleotides labeled in grey indicate random nucleotides not complementing the full sequence of mature miR-1; nucleotides labeled in yellow and red indicate nucleotides in the miRNA target sequence that are complementary and non-complementary, respectively, to the seed sequence; nucleotides labeled in light-green indicate the sequence differences in the miR-124 sequence. (**B**) Schematic illustration of the dGFP Fluorescent Recovery After Photobleaching (FRAP) assay for measuring rates of protein synthesis. At 24-h after transfection, GFP fluorescence in cells is photobleached; the build-up of the fluorescent signal is then recorded for 10 min. Time-lapse movies are analyzed to calculate the percent of fluorescent signal build-up compared to the baseline signal after photobleaching. (**C**) Representative images of dGFP in 293T cells carrying miR-1-3p canonical sites in 3′UTR co-transfected with mature miR-1-3p. Images show three time points for each cell: 0, 5, and 10 min after photobleaching. (**D**) A graph showing the cumulative percent of fluorescence of dGFP during the 10 min following photobleaching. The horizontal K_R50_ line indicates the Y value at which 50% of the maximal recovery is achieved in the non-targeting control (miR-124-cy3). K_R50_ is used as a reference point for comparing kinetic features of protein synthesis and is marked with a vertical colored dashed line for every condition. (**E**) Quantification of the mean slope of dGFP recovery after photobleaching. Ordinary one-way ANOVA **** *p* < 0.0001. Post-hoc multiple comparisons: ** *p*-value < 0.01, **** *p*-value < 0.0001. (**F**) NGS quantification of the transcript levels of miR-1-3p targets categorized by seed sequence at 0 h, 2 h, and 4 h after miR-1-3p transfection into 293T cells. ** *p* < 0.01 (**G**) Real-time PCR quantification of the relative mRNA expression of three genes representative of the seed sequence group, at 4 h after transfection with miR-1-3p or with scrambled RNA, compared to 0 h after transfection. * *p* < 0.05, ** *p* < 0.01.

**Figure 2 cells-12-00233-f002:**
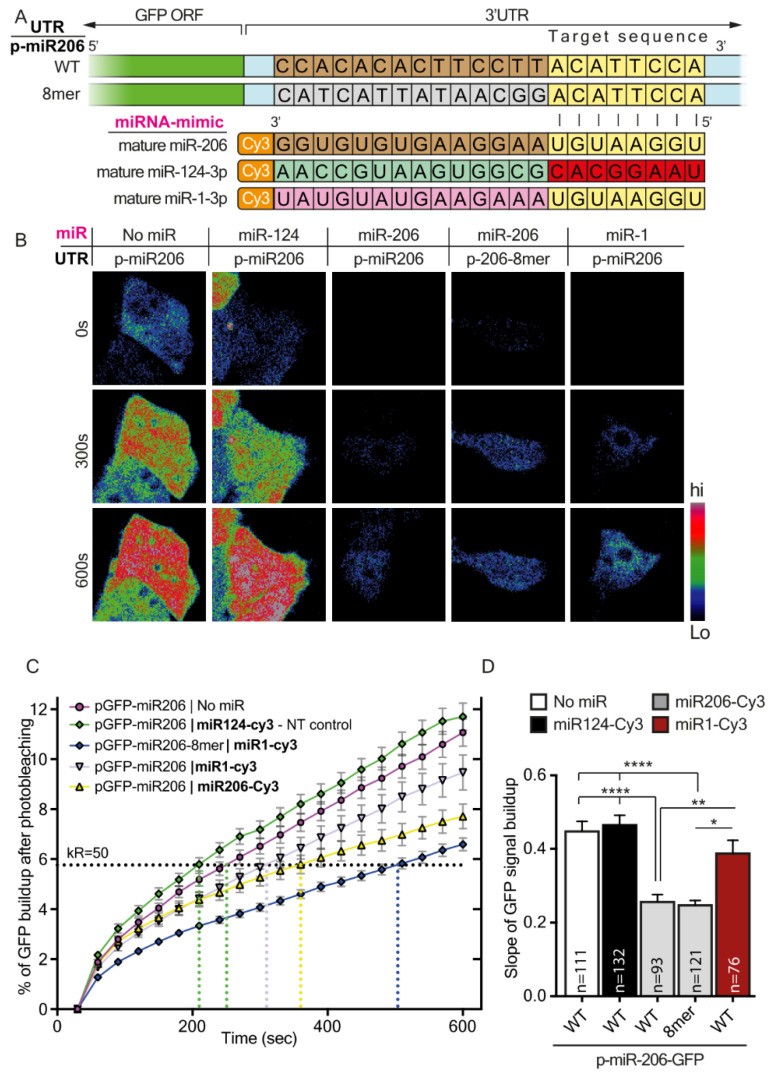
Target GFP downregulation as a function of canonical site variations applies also for miR-206-3p. (**A**) Schematic illustration of the experimental setup. Two dGFP expression plasmids containing miR-206-3p canonical sites in the 3′UTR of dGFP were each co-transfected together with mature Cy3-miR-206 or mature Cy3-miR-1, or mature Cy3-miR-124 as a control, into 293T cells. Green indicates the GFP coding sequence (CDS); light blue indicates the 3′UTR of dGFP; nucleotides labeled in brown indicate nucleotides in full complementation to mature miRNA; nucleotides labeled in grey indicate random nucleotides not complementing the full sequence of mature miR-1; nucleotides labeled in yellow and red indicate nucleotides in the miRNA target sequence that are complementary and non-complementary, respectively, to the seed sequence; nucleotides labeled in light-green and pink highlight the sequence differences between mature miR-206, miR-124, and miR-1. (**B**) Representative images of dGFP in 293T cells carrying miR-206 canonical sites in 3′UTR. The images show three time points for each cell: 0 min, 5 min, and 10 min after photobleaching. (**C**) The graph shows the cumulative percent of fluorescence of dGFP during the 10 min following photobleaching. The horizontal K_R50_ line indicates the Y value at which 50% of the maximal recovery is achieved in the non-targeting control. (**D**) Quantification of the mean slope of dGFP recovery after photobleaching. Ordinary one-way ANOVA **** *p* < 0.0001. Post-hoc multiple comparisons: * *p*-value < 0.05, ** *p*-value < 0.01, **** *p*-value < 0.0001.

**Figure 3 cells-12-00233-f003:**
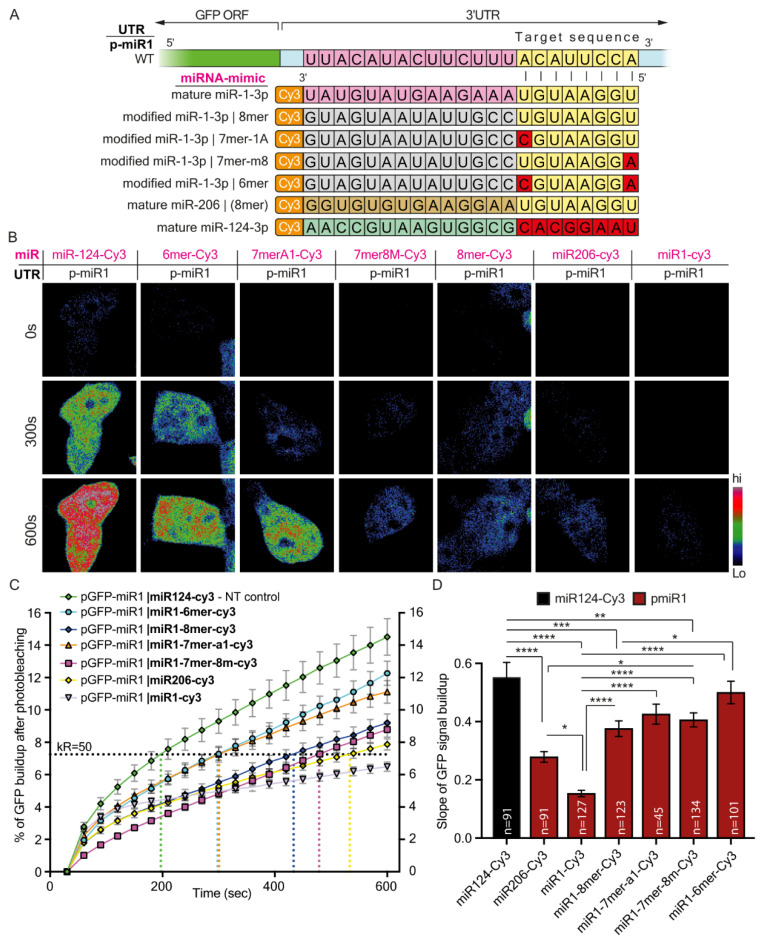
Modified miR-1-3p mimetics differentially regulate protein synthesis of WT 3′UTR. (**A**) A schematic illustration of the experimental setup. Four distinct miRNA mimetics containing mutations in their seed sequence, which mimic the canonical sites usually found on target 3′UTR, and three more mature miRNAs (Cy1-miR-1, Cy3-miR-206, Cy3-miR-124) were each co-transfected into 293T cells, together with dGFP expression plasmid containing the full complementary sequence for miR-1-3p. Green indicates the GFP coding sequence (CDS); light blue indicates the 3′UTR of dGFP; nucleotides labeled in pink indicate nucleotides in full complementation to mature miR-1; nucleotides labeled in grey indicate random nucleotides not complementing the full sequence of mature miR-1, miR-206,and miR-124; nucleotides labeled in yellow and red indicate nucleotides in the miRNA target sequence that are complementary and non-complementary, respectively, to the seed sequence; nucleotides labeled in light green and brown indicate sequence differences between miR-124, miR-206, and miR-1. (**B**) Representative images of dGFP in 293T cells carrying WT miR-1-3p canonical sites in 3′UTR, but co-transfected with modified miR-1-3p variants. The images show three time points for each cell: 0 min, 5 min, and 10 min after photobleaching. (**C**) A graph showing the cumulative percent of fluorescence of dGFP during the 10 min following photobleaching. The horizontal K_R50_ line indicates the Y value at which 50% of maximal recovery is achieved in the non-targeting control. K_R50_ is used as a reference point for comparing kinetic features of protein synthesis and is marked with a vertical-colored dashed line for every condition (**D**) Quantification of the mean slope of dGFP recovery after photobleaching. Ordinary one-way ANOVA **** *p* < 0.0001. Post-hoc multiple comparisons: * *p*-value < 0.05, ** *p*-value < 0.01, *** *p*-value < 0.001, **** *p*-value < 0.0001.

**Figure 4 cells-12-00233-f004:**
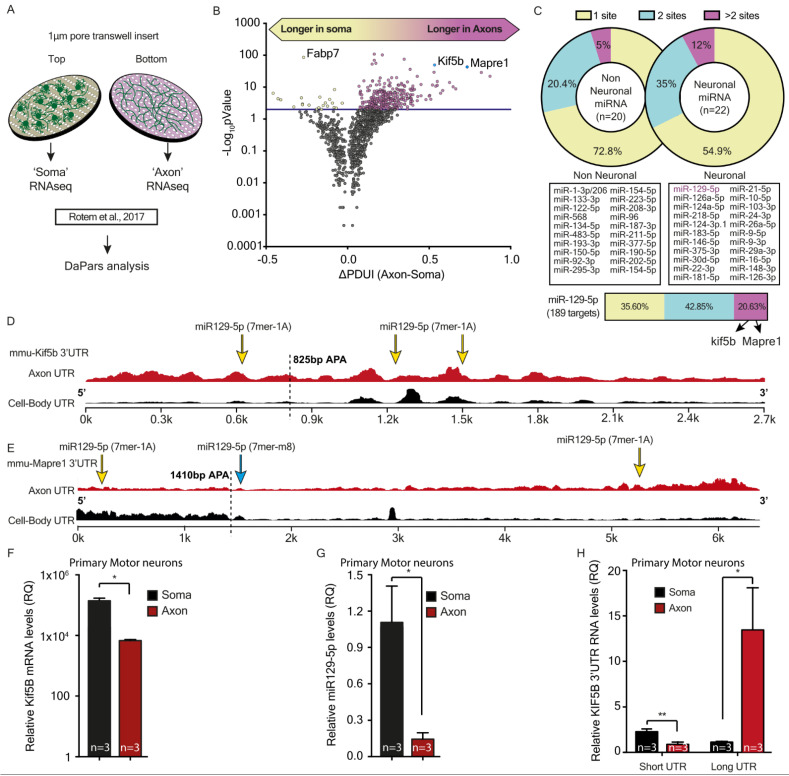
Kif5b mRNA transcript variant length and expression at axon versus cell-body motor neurons. (**A**) Illustration of the transwell insert method used for isolation of axonal RNA from primary motor neuron (MNs). (**B**) A volcano plot demonstrating results of the Dynamitic analysis of Alternative Poly-Adenylation from RNA-seq (DaPars) analysis for the ΔPDUI, representing the differences in the short vs. long 3′UTR ratios in axons vs. cell bodies. Significantly elongated 477 mRNA transcripts are colored in red. Kif5b and Mapre1 (blue) were chosen for further validations. (**C**) Upper panel: analysis of the percent of long 3′UTR transcripts containing 1, 2, or >2 target sequences for neuronal versus non-neuronal miRNAs. The middle panel shows the list of neuronal and non-neuronal miRNAs tested. The lower panel details the specific percent of miR-129-5p targets within the long 3′UTR transcripts that contain 1, 2, or >2 target sequences. Two transcripts with more than two target sequences for miR-129-5p are highlighted: Kif5b and Mapre1. (**D**) The diagram compares 3′UTR coverage of the Kif5b transcript in axons (red) versus cell-bodies (black). The alternative polyadenylation (APA) site is marked with a black dashed line. The yellow arrows indicate positions of miR-129-5p target sequences. (**E**) The diagram compares 3′UTR coverage of Mapre1 transcript in axons (red) versus cell-bodies (black). The APA site is marked with a black dashed line. The yellow arrows indicate positions of miR-129-5p target sequences. (**F**–**H**) RT-qPCR analysis of (**F**) Kif5b coding sequence, (**G**) mature miR-129-5p, or (**H**) shortened and elongated Kif5b 3′UTR expression in axons and cell bodies of primary MNs. Normalized fold change (FC) is shown in Kif5b and miR-129-5p expression to GAPDH and miR-124a-3p, respectively. The 3′UTR expression in each compartment (**H**) is relative to the expression of the coding sequence exons. Unpaired students’ t-test. * *p* < 0.05, ** *p* < 0.01.

**Figure 5 cells-12-00233-f005:**
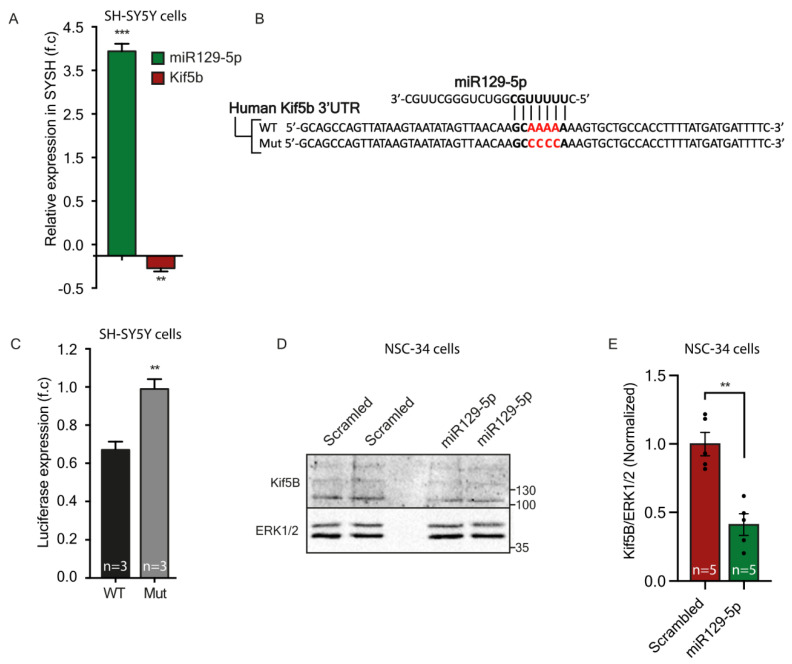
miR-129-5p targets Kif5b mRNA and regulates Kif5b protein levels. (**A**) Real-time PCR analysis of normalized miR-129-5p and Kif5b mRNA expression following 24 h of miRNA overexpression (OE) in the SH-S5Y5 neuroblastoma cell line, relative to control plasmid OE. Unpaired students’ t-test. ** *p* < 0.01, *** *p* < 0.0001. (**B**) Sequences of Renilla/Firefly luciferase under the regulation of Kif5b 3′UTRs that were used for transient reporter assay experiments. Wild-type (WT) and mutant (mut) alleles for miR-129-5p binding sites are presented. The miRNA seed region and complimentary 3′ UTR sequence are marked in bold. Mutagenic nucleotides are marked in red and bold. (**C**) Luciferase activity 48 h following co-transfection of miR-129-5p with Renilla/Firefly luciferase constructs under the regulation of Kif5b 3′UTR. The graph shows the relative expression level of Firefly luciferase standardized to Renilla luciferase. The paired two-tail Student’s t-test was used for statistical analysis (n = 3, ** *p* < 0.01). (**D**,**E**) Western blot (**D**) and analysis of Kif5b protein (~110 kDa) levels in NSC-34 overexpressing miR-129-5p or scrambled control for 72 h. (**E**). ERK1/2 (42–44 kDa) was used as a loading control. ** *p* < 0.01 student’s t-test, n = 5 independent experiments.

**Figure 6 cells-12-00233-f006:**
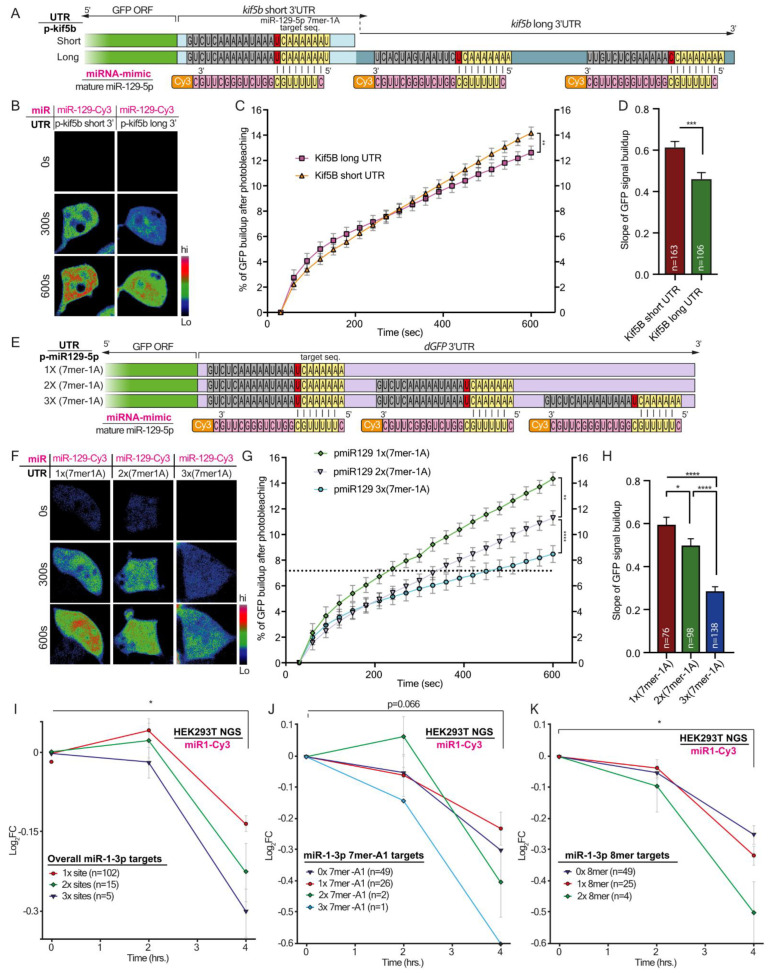
The number of miR-129-5p binding sites differentially regulates the 3′UTR variants of Kif5b and dGFP. (**A**) A schematic illustration of the experimental setup. The 3′UTR of dGFP was replaced with the 3′UTR of either short or long isoforms of Kif5b. Green indicates dGFP coding sequence (CDS). Light and dark blue indicate short and long 3′UTRs Kif5b, respectively. Nucleotides labeled with grey do not complement the mature miR-129-5p sequence. Nucleotides labeled with yellow represent the 7mer-A1 target sequence for miR-129-5p seed. Nucleotides labeled in pink are part of mature miR-129-5p (**B**) Representative images of dGFP in 293T cells carrying short or long 3′UTR of Kif5b, fused to dGFP co-transfected with mature miR-129-5p. Three time points are shown for each cell: 0 min, 5 min, and 10 min after photobleaching. (**C**) The graph shows the cumulative percent of fluorescence of dGFP during the 10 min following photobleaching. ** *p*-value < 0.01. (**D**) Quantification of the mean slope of dGFP signal build-up after photobleaching. *** *p*-value < 0. 001. (**E**) A schematic illustration of the experimental setup. The 3′UTR of dGFP was modified to contain 1X, 2X, or 3X 7mer-A1 target sites for miR-129-5p. Green indicates the dGFP coding sequence (CDS). Light magenta indicates the unmodified 3′UTR of dGFP. Nucleotides labeled with grey do not complement the mature miR-129-5p sequence. Nucleotides labeled with yellow represent the 7mer-A1 target sequence for miR-129-5p seed. Nucleotides labeled in pink are part of mature miR-129-5p (**F**) Representative images of dGFP in 293T cells carrying single, double, or triple miR-129-5p binding sites in the 3′UTR of dGFP co-transfected with mature miR-129-5p. The images show three time points for each cell: 0 min, 5 min, and 10 min after photobleaching. (**G**) The graph shows the cumulative percent of fluorescence of dGFP during the 10 min following photobleaching. ** *p*-value < 0.01, **** *p*-value < 0.0001 (**H**) Quantification of the mean slope of dGFP signal build-up after photobleaching. Ordinary one-way ANOVA **** *p* < 0.0001. Post-hoc multiple comparisons: * *p*-value < 0.05, **** *p*-value < 0.0001. (**I**–**K**) NGS quantification of the transcript levels of the miR-1-3p targets categorized by the (**I**) overall number of sites, * *p*-value < 0.05, (**J**) the number of 7mer-A1 sites, and (**K**) 8mer binding sites for miR-1-3p at 0 h, 2 h, and 4 h after miR-1-3p transfection into 293T cells. * *p*-value < 0.05.

**Figure 7 cells-12-00233-f007:**
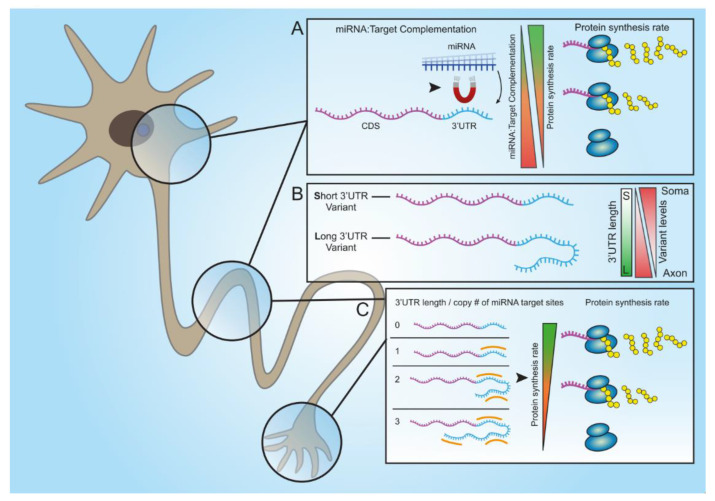
A model of suggested pathways for miRNA-based post-transcriptional regulation at subcellular resolution within motor neuron axons. (**A**) The efficacy of miRNA silencing in cells is determined by the target sequence complementation of the miRNA. Higher complementation (e.g., 8mer) yields stronger target silencing, which results in slower protein synthesis. (**B**) Axonal mRNA transcripts contain longer 3′UTR than do their cell-body variants. (**C**) The long 3′UTRs of axonal transcripts contain multiple miRNA binding site repeats for specific miRNA, which can differentially regulate their subcellular translation. A high repeat number correlates with a lower rate of target translation.

**Table 1 cells-12-00233-t001:** Mature miRNAs or 3′UTR canonical site sequences.

Primers/miRNAs	Sequence and Dye
miR-1-3p-CY3	5′-UGGAAUGUAAAGAAGUAUGUAU-CY3-3′
miR-1-3p full complementation 3′UTR (WT)	5′-TTACATACTTCTTTACATTCCA-3′
miR-1-3p 8mer 3′UTR	5′-CATCATTATAACGGACATTCCA-3′
miR-1-3p 7mer-8m 3′UTR	5′-CATCATTATAACGGACATTCCT-3′
miR-1-3p 7mer-1A 3′UTR	5′-CATCATTATAACGGGCATTCCA-3′
miR-1-3p 6mer 3′UTR	5′-CATCATTATAACGGGCATTCCT-3′
miR-124-3p-CY3	5′-UAAGGCACGCGGUGAAUGCCAA-CY3-3′
miR-206-CY3	5′-UGGAAUGUAAGGAAGUGUGUGG-CY3-3′
miR-206 full complementation 3′UTR	5′-CCACACACTTCCTTACATTCCA-3′
miR-206 8mer 3′UTR	5′-AATACTTAGAAAGGACATTCCA-3′
miR-1-3p-8mer-CY3	5′-UGGAAUGUCCGUUAUAAUGAUG-CY3-3′
miR-1-3p-7mer-8m-CY3	5′-AGGAAUGUCCGUUAUAAUGAUG-CY3-3′
miR-1-3p-7mer-1A-CY3	5′-UGGAAUGCCCGUUAUAAUGAUG-CY3-3′
miR-1-3p-6mer-CY3	5′-AGGAAUGCCCGUUAUAAUGAUG -CY3-3′
miR-129-5p-CY3	5′-CUUUUGCGGUCUGGGCUUGC -CY3- 3′
miR-129-5p 7mer-1A 3′UTR	5′-TCTCAAAAATAAATCAAAAAA-3′

**Table 2 cells-12-00233-t002:** SYBR green RT-PCR primers for mRNA quantification.

Primer.	Sequence
hsa-Kif5b For	CATACAATGAGTCTGAAACAAAATCTACACTCTTATT
hsa-Kif5b Rev	CACAAACTGTGTTCTTAATTGTTTTGGC
hsa-GAPDH for	CCACTCCTCCACCTTTGACGCT
hsa-GAPDH rev	ACCCTGTTGCTGTAGCCAAATTCG
mmu-Kif5b CDS For	TAACCTTTCAGTCCATGAAGACAAAAACC
mmu-Kif5b CDS Rev	GACTTCATCTGGACTACACACGAAACG
mmu-Gapdh For	GAGTATGTCGTGGAGTCTACTGGTGTCTTC
mmu-Gapdh Rev	CGGAGATGATGACCCTTTTGGCT
Short 3′UTR For	GGTGTGTCCTTCGTGTCTTCACTGT
Short 3′UTR Rev	TTAGTAGAAAAGGGAAAATGAAAAGCAATAGC
Long 3′UTR For	GATAATTGGTTCAGAAGAGAAACTCAATGAAA
Long 3′UTR Rev	TAGACTCTCCTCTGTTACCTCAAATCAAACTG

**Table 3 cells-12-00233-t003:** TaqMan RT-PCR probe and primer assay ID for miRNA quantification.

miRNA	Thermo Fisher Scientific Assay ID
miR-129-5p	000590
miR-124a-3p	000446
miR-1-3p	002222
miR-206	000510

**Table 4 cells-12-00233-t004:** Descriptive statistics for Figure 1D—modifications in p-miR1-GFP target sequences.

	miR1-124	miR1-6mer	miR1-8mer	miR1-7mer-A1	miR 1 -7mer 8m	pmiR1-miR1
**p-miR1|miR1-124-cy3**		0.18723866	0.00140018	0.00318815	0.01248199	8.1725E-14
**p-miR1-6mer**	0.18723866		0.01491379	0.01201426	0.04581773	5.8869E-13
**p-miR1-8mer**	0.00140018	0.01491379		0.27336066	0.49944018	9.0681E-17
**p-miR1-7mer-A1**	0.00318815	0.01201426	0.27336066		0.32428856	6.5754E-06
**p-miR 1 -7mer 8m**	0.01248199	0.04581773	0.49944018	0.32428856		2.1994E-07
**p-miR1|miR1-cy3**	8.1725E-14	5.8869E-13	9.0681E-17	6.5754E-06	2.1994E-07	

**Table 5 cells-12-00233-t005:** Descriptive statistics for Figure 2C—modifications in p-miR206-GFP target sequences.

	miR-124	miR-206	8mer	GFP (miR-1)	GFP only
**p-miR206|miR124-cy3**		2.23E-07	4.185E-15	1.08E-07	0.209767
**p-miR206|miR206-cy3**	2.23E-07		0.0158373	0.395211	5.75E-06
**p-miR206-8mer**	4.19E-15	0.015837		0.001321	2.75E-13
**p-miR206|miR1-cy3**	1.08E-07	0.395211	0.0013213		2.2E-06
**p-miR206**	0.209767	5.75E-06	2.746E-13	2.2E-06	

**Table 6 cells-12-00233-t006:** Descriptive statistics for Figure 2C—modifications in synthetic miRNA-1.

	miR124-Cy3	miR1-Cy3	8mer-Cy3	6mer-Cy3	7mer8m-Cy3	7mer1A-Cy3	miR206-cy3
**miR124-Cy3**		8.173E-14	4.806E-06	0.046807	2.083E-07	0.0234587	0.0002063
**miR1-Cy3**	8.173E-14		9.497E-06	9.615E-14	2.381E-05	5.69E-12	4.508E-06
**8mer-Cy3**	4.806E-06	9.497E-06		0.0004608	0.2907446	0.029917	0.379676
**6mer-Cy3**	0.046807	9.615E-14	0.0004608		2.738E-05	0.1732727	0.0043319
**7mer8m-Cy3**	2.083E-07	2.381E-05	0.2907446	2.738E-05		0.0056587	0.204547
**7mer1A-Cy3**	0.0234587	5.69E-12	0.029917	0.1732727	0.0056587		0.0608088
**miR206-cy3**	0.0002063	4.508E-06	0.379676	0.0043319	0.204547	0.0608088	

## Data Availability

Raw data can be accessed via the NCBI GEO repository using accession number GSE188770.

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
