# Peer review of "Multiple Copies of microRNA Binding Sites in Long 3′UTR Variants Regulate Axonal Translation"

_cells, 2023, doi:10.3390/cells12020233_

Round 1
Reviewer 1 Report
Please see the attached word file as formatting was corrupted if I copied directly from there.

Author Response
We thank the reviewers for their general interest in our work and their constructive input. We have addressed the comments of each reviewer point-by-point in the attached file. We hope our responses to the points raised will satisfy the reviewers and that our revised manuscript will be found suitable for publication.

Reviewer 2 Report
In this research article, Farberov et al. used a series of constructs encoding miRNA and targeted RNA to investigate the relation between miRNA translational inhibitory efficiency and miRNA-seed region sequence matching or the number of binding sites located at the mRNA 3’UTR region. While all FRAP assays to measure translation rates were carefully designed and analysed, the overall experimental evidence included in the manuscript is less satisfactory to propose a mechanism of how axonal miRNA regulates axonal mRNA translation differently from other neuronal compartments or other cell types.
The main pitfall of this study is the lack of neuronal evidence showing the same mechanism of miRNA inhibition observed in cell bodies or other cell types applies to axons. For example, all experiments regarding the seed sequence variants were performed in HEK. While the authors have convincingly shown that the inhibitory effect of miRNAs on their target mRNA translation was compromised as mismatched bases increased using a collection of mutant constructs, the reviewer failed to see the connection between Fig. 1-3 and the rest (and the main argument) of the paper - axonal translation regulated by miRNA. As the analysis of genome-wide transcript 3’UTRs identified extremely rare axonal miRNA seed area variants, the mechanism proposed in the first half of the paper cannot explain the main differences between somal and axonal miRNA regulation. However, the authors may wish to investigate if such fine-tuning of base pairing between miRNA and seed regions is a global mechanism to regulate miRNA efficiency of translational inhibition in neuronal soma by examining seed region variant frequency within the neuronal soma mRNA 3’UTR. If proven to be true, it can also be an interesting soma-axon difference of translational regulation by miRNA.
The authors argue that kif5b is axonally enriched, but all the subsequent tests of its regulation by miR-129 were performed in HEK cells. As Fig. 4F and 4G show, the availability of miR-129 and kif5b mRNA between axon and soma is different. The study did not show the direct consequence of longer 3’UTR of axonal kif5b. Do the authors try to argue that the axonal kif5b is more translationally repressed due to the larger number of miRNA binding sites? If so, evidence should be provided with neuronal data (axon vs. soma) to support this claim.
Work from the Marie-Laure Baudet lab on miRNA regulating local translation in axons, which is highly relevant to this study, should be acknowledged. For instance, a list of axonally enriched miRNA was summarised in this review from the Baudet lab (Corradi and Baudet, 2020) and a few reviews cited in the introduction. Please discuss if any of the miRNAs investigated in the current study has previously shown to be axonally enriched. If not, can the authors provide experimental evidence that these miRNAs are indeed axonally localised?
Reference
Corradi, E., and Baudet, M.L. (2020). In the Right Place at the Right Time: miRNAs as Key Regulators in Developing Axons. Int J Mol Sci 21. 10.3390/ijms21228726.
Author Response

(The authors gave the same response as above.)

Round 2
Reviewer 2 Report
The revised manuscript has addressed most of the reviewer's comments and has improved with the additional results and discussions. Given the limited time and technical challenges to perform axonal FRAP experiments, the reviewer agrees that the current manuscript is suitable for publication in Cells.